# Body-Size Misperception among Overweight Children and Adolescents in Greece: A Cross-Sectional Study

**DOI:** 10.3390/nu15081814

**Published:** 2023-04-08

**Authors:** Panagiotis Plotas, Efstathia Tsekoura, Emmanouil Souris, Anastasios Kantanis, Eirini Kostopoulou, Anastasia Varvarigou, Sotirios Fouzas

**Affiliations:** 1Laboratory Primary Health Care, School of Health Rehabilitation Sciences, University of Patras, 26504 Patras, Greece; 2Department of Pediatrics, School of Medicine, University of Patras, 26500 Patras, Greece

**Keywords:** childhood obesity, child and adolescent development, diet, eating disorders/body image, family health, health promotion, obesity, pediatrics, public health

## Abstract

Childhood obesity can affect both physical and mental health. Body-size misperception may lead to a lack of motivation to make healthy changes or to engage in unhealthy weight loss behaviors, increasing the possibility for obese children to become obese adults. To estimate the frequency of body-size misperception among children and adolescents, we conducted a cross-sectional study within another study on eating disorders in youth in Greece (National Institute of Educational Policy, act no. 04/2018). Between January and December 2019, two trained assistants visited 83 primary and secondary schools of the Region of Western Greece and interviewed 3504 children aged 10–16 years (CL 99%) and performed anthropometric measurements. Among the 3504 surveyed children, 1097 were overweight, including 424 obese, and 51 were underweight. The “perceived” BMI was not computed in 875 children (25%), who did not state their weight or height and were classified as non-responders. Weight bias was inversely related to BMI, the obese and overweight non-obese children underestimated their weight, while the underweight children overestimated it. Conversely, height bias was positively related to BMI bias. BMI bias was not related to sex, age, parental education, or place of residence. In conclusion, our study lends robust support to the existing evidence on unrealistic body images among overweight children and adolescents. Prompt recognition of such misperceptions may help in increasing motivation towards healthier eating habits, systematic physical activity, and weight-control interventions.

## 1. Introduction

Body-size misperception refers to when an individual inaccurately perceives their own weight status. This can occur among overweight children and adolescents, where they may perceive themselves as being smaller or thinner than they actually are [1,2,3]. Self-perceived body image is subjective and may be influenced by family and cultural stereotypes, media and social comparisons [4]. People may feel pressure to conform to specific body types and sizes that are considered attractive or desirable in their community or culture. This can lead to negative self-esteem, body dissatisfaction, and a distorted view of one’s body. For example, some cultures may promote a thin and slender body type, while others may emphasize a curvy and voluptuous figure. These societal pressures can cause individuals to view their bodies as inadequate or unattractive, leading to negative self-perceptions.

It is important for healthcare professionals to assess for body-size misperception when working with overweight children and adolescents and to provide education and support to help them accurately perceive their weight status. Assessing for body-size misperception involves evaluating an individual’s perception of their body size and comparing it to their actual body size and composition. This can be carried out through measurements such as body mass index (BMI), waist circumference, and body fat percentage, as well as through subjective assessments such as questionnaires and interviews. When working with overweight children and adolescents, it is crucial for healthcare professionals to approach the topic of body-size misperception with sensitivity and empathy [5]. A negative body image can lead to low self-esteem, anxiety, and depression, and it is essential to address these concerns in a supportive and non-judgmental manner [6].

Child obesity can lead to a variety of serious health problems, such as diabetes, heart disease, and high blood pressure, and obese children are more likely to become obese adults, which can lead to a lifetime of health complications [7]. Additionally, childhood obesity can have a negative impact on mental and emotional well-being and can lead to social and psychological problems [7].

Child obesity is a growing concern in Greece, as the number of overweight and obese children has increased in recent years. According to data from the World Health Organization (WHO), in 2016, over 25% of Greek children aged 5–17 were overweight or obese, and Greece has the highest rates of childhood obesity among the countries in the WHO European Region [8]. Several factors contribute to the high rates of child obesity in Greece, including poor dietary habits (skipping breakfast) [9], lack of physical activity, and sedentary lifestyle [10]. Many children consume high-calorie and high-fat foods, and there is limited access to fresh fruits and vegetables in some areas [11].

Inactivity is also a significant issue, as many children spend long hours in front of screens, such as televisions and computers. This lack of physical activity, combined with poor dietary habits, leads to an increase in body weight and a higher risk of obesity-related health problems [12]. Healthcare professionals can also play a critical role in promoting healthy behaviors and body image by educating children and adolescents on the importance of healthy eating habits, physical activity, and self-care. Encouraging them to adopt positive self-talk and to focus on their strengths and abilities can also help to boost their self-esteem and promote a healthy body image.

BMI bias results in a lack of motivation to change one’s diet or physical activity as individuals may not realize the need to make healthy changes to their lifestyle to maintain a healthy weight and body composition. Between January and December 2019, we conducted the first survey on body-size misperceptions among children and adolescents in Greece. In our survey, we explored the agreement between actual and self-reported weight and height as a more objective measure of body-size perception than generic questions, in a large cohort of Greek children and adolescents from primary and secondary schools of the Region of Western Greece.

## 2. Materials and Methods

This survey was nested within a study on eating disorders in youth in Greece (National Institute of Educational Policy, act no. 04/2018) and was approved by the Ethics Committee of the University of Patras. Written parental consent and verbal participants’ assent were obtained. The study included 3504 children and adolescents aged 10–16 years (CL 99% ± CI 5%) from 83 primary and secondary schools of the Region of Western Greece (15% of regional educational institutions), using convenience sampling. The only exclusion criterion was the lack of parental consent or verbal participant’s assent.

Between January and December 2019, two trained assistants from the University Hospital of Patras visited the schools, interviewed the children and performed anthropometric measurements. During the interview, participants were asked to state their weight and height, being unaware that anthropometric measurements would follow in order to minimize socially desirable answers to please the researcher. All children were examined dressed the same way. More specifically, body weight was measured with the least possible clothing to the nearest 0.1 kg using a Seca 877 scale, while height was measured without shoes to the nearest 0.1 cm with a Seca 217 stadiometer. The body mass index (BMI) was calculated as height/weigh^2^. BMI z-scores and percentiles were determined according to extended international cut-offs in terms of the underlying LMS curves [12]. 

Weight, height, and BMI bias were defined as the differences between reported and measured values. One-way ANOVA with Bonferroni post hoc test was performed for all between-group differences. We used chi-square test to examine non-responder rates across the BMI strata and the effect of sex, age, parental education, and place of residence. BMI z-scores were assessed with univariable and multivariable linear regression. All analyses were performed with SPSS v.28 (IBM Corp., Armonk, NY, USA). 

## 3. Results

Of the 3504 surveyed children (Table 1), 1097 (31.3%) were overweight, including 424 (12.1%) obese, and 51 (1.5%) were underweight. The “perceived” BMI was not computed in 875 children (25%) who did not state their weight (N = 821, 23.4%) or height (N = 735, 21%) and were classified as non-responders. The population characteristics did not differ between responders and non-responders.

The weight bias was inversely related to BMI (linear regression analysis; Figure 1); the obese and overweight non-obese children underestimated their weight (bias −10 ± 5.6 kg and −4.3 ± 4.3 kg, respectively). Conversely, the height bias was positively related to BMI (Figure 1). As a result, the “perceived” BMI was inversely related to actual BMI (Figure 1); the obese and overweight participants underestimated their BMI (bias −3.8 ± 2.4 kg/m^2^ and −1.7 ± 2 kg/m^2^, respectively) while the underweight children overestimated it (bias 0.7 ± 1 kg/m^2^). All between-group differences were significant (one-way ANOVA with Bonferroni post hoc test; *p* < 0.001). Non-responder rates were comparable across the BMI strata (chi-square test; Figure 1). The BMI bias was solely dependent on BMI z-scores (multivariable regression coefficient β −0.456; *p* < 0.001), and not related to sex (β 0.022; *p* = 0.271), age (β −0.030; *p* = 0.130), parental education (β −0.015; *p* = 0.462) or place of residence (β 0.027; *p* = 0.184).

## 4. Discussion

We found that overweight children and adolescents in Greece not only underestimate their weight, but also tend to overestimate their height. Thus, the BMI calculated from self-reported weight and height values is in significant disagreement with their actual BMI. This inverse relationship between perceived and actual body size was present across the whole BMI spectrum. The relationship was not influenced by sex, family education level or urban, semi-urban, or rural living areas of the participants.

Previous studies have shown that 75% of overweight children in the USA [3], 43% in Europe [1] and 25% in China [2] believed that their weight was ideal. The magnitude of body size underestimation is variable and depends on various ethnic and socio-demographic parameters [1,2,3,13,14]. However, a possible limitation of these studies could be that body image perception was assessed through generic questions like “do you consider yourself normal weight, overweight or thin?” [1,2,3]. Apparently, the responses in such questions can be subjective and influenced by various factors such as societal and cultural norms, media, and personal beliefs.

In our study, we asked children to report their weight and height, and we used these values to calculate the “perceived” BMI as a more robust measure of self-perceived body image. This type of approach provides a more accurate and nuanced understanding of an individual’s body perception. It is highly unlikely that our participants recalled lower weight values while reporting accurate or even higher values for the height. However, it is possible that some overweight participants provided more “socially acceptable” answers. 

Among the participants, overweight and obese children and adolescents had a larger gap between “perceived” and actual BMI than normal or underweighted children. This observation is in agreement with the results of a study by Bordeleau et al. [15] and may occur because of these children’s heightened need to conform to social norms. Furthermore, previous studies have found a relationship between body-size misperception and socio-demographic characteristics of the participants. Sarafrazi et al. [3] found that BMI bias was lower among children and adolescents from higher income families compared with those from middle- or lower-income families, and in our study parental education or place of residence were not related. Furthermore, in previous studies, weight status misperception was more common among boys than girls [2,3]. This may occur due to the social desirability for a lower body weight in girls and for a larger body size in boys. This can ultimately lead to correctly perceive an overweight girl, but simultaneously to misperceive an overweight boy as an early developed, normal-weight child. However, in our study, body-size misperception was not related to sex and age. 

The lack of correlation between socio-demographic characteristics and body-size misperception observed in our study may be due to the general characteristics of Greek families and society as well as the high prevalence of obesity in the entire child population in our country. In a large national survey polled in 2015, Tambalis et al. studied the prevalence of obesity in 336,014 (51% boys) children aged 4 to 17 years old from almost 40% of all schools of primary and secondary education in Greece [10]. They found no statistically significant relationship between income, education or place of residence and obesity prevalence. Despite that, age, physical fitness, low adherence to the Mediterranean diet, insufficient sleeping hours, inadequate physical activity levels and increased screen time were all associated with higher odds of total and central obesity. 

Another final important outcome from our survey is that underweighted children and adolescents overestimated their BMI. In addition to obesity and the problems created by BMI bias in potential interventions, the overestimation of weight in underweight children also raises concerns as it can lead to further negative impacts on children’s health and well-being [16]. Children who believe they are overweight, when in reality they are underweight, may reduce their food intake, leading to further weight loss and reduced growth and development, increase the risk of anemia due to iron deficiency and experience delayed puberty and related developmental issues. On top of that, they may experience increased psychological distress, leading to depression, anxiety and anorexia. Finally, a lack of energy and muscle mass may decrease their physical performance. It is important for healthcare providers to provide nutrition education and promote healthy habits in order to assess and correct any body size misperception in underweight children.

## 5. Strengths and Limitations

Our study has several strengths. In contrast to previous studies related to obesity, it is homogeneous, has a wide age range and a clear target group. We also used a standard protocol to measure anthropometric data and the fact that all measures were conducted by the same two trained assistants may have minimized the inter-observer measurement error.

A possible limitation in our study is that BMI does not consider muscle mass, bone density, and distribution of body fat. Therefore, athletic children with high muscle mass will be erroneously classified as over-weight or obese, overestimating their BMI. A further limitation is that children were interviewed without the presence of their parents, and they may not feel comfortable or safe. Furthermore, they may have given socially desirable answers to please the researcher.

## 6. Conclusions

In conclusion, despite limitations, this is the first study on body-size misperception among children and adolescents in Greece. Our survey lends robust support to the existing evidence on unrealistic body images among overweight children and adolescents [1,2,3]. According to our findings, overweight children and adolescents in Greece not only underestimate their weight, but also tend to overestimate their height, and the BMI calculated from self-reported weight and height values is in significant incompatibility with their actual BMI. Body size misperception among children and adolescents can have a significant impact on the lack of motivation for lifestyle changes to improve health and leads to a continuation of unhealthy habits and an increased risk of obesity-related health problems. Additionally, if healthcare providers are unable to effectively communicate the need for weight loss due to a patient’s misperception of their body size, it can hinder the ability to provide proper treatment and care. Greece has the highest rates of childhood obesity among European countries; thus, prompt recognition of such misperceptions is critical and may help in increasing motivation towards healthier eating habits, systematic physical activity, and weight-control interventions.

Population characteristics are shown in the Appendix A. The relationship between BMI z-score and weight, height, and BMI bias are shown in Figure 1. The non-responder rate per BMI strata and the BMI z-score ranges for overweight and obesity are also presented in Figure 1.

## Figures and Tables

**Figure 1 nutrients-15-01814-f001:**
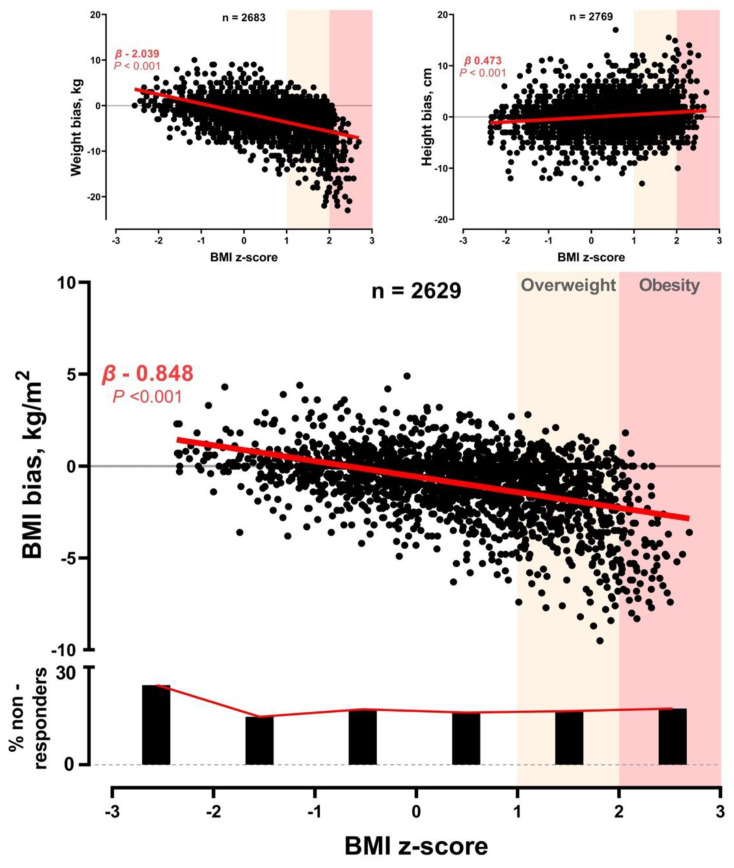
Relationship between BMI z-score and weigh, height, and BMI bias. Negative values indicate underestimation while positive values indicate overestimation of the respective parameter. The non-responder rate per BMI strata and the BMI z-score ranges for overweight and obesity are also presented. BMI: body mass index. β: unstandardized coefficients beta (linear regression).

**Table 1 nutrients-15-01814-t001:** Characteristics of the study sample.

Characteristics	Males	Females	All
Surveyed, no. (%)	1759	1745	3504
Age, mean ± SD (IQR), y	12.8 ± 1.4 (11.6–14)	12.9 ± 1.5 (11.6–14.1)	12.8 ± 1.4 (11.6–14)
Residence			
Urban, ^a^ no. (%)	1126 (64)	1169 (67)	2295 (65.5)
Rural, no. (%)	633 (36)	576 (33)	1209 (34.5)
Parental education score, ^b^ mean ± SD (IQR)	5.3 ± 1.9 (4–7)	5.5 ± 1.6 (4–7)	5.4 ± 1.8 (4–7)
Number of brothers/sisters, mean ± SD (IQR)	1.5 ± 1.2 (2–3)	1.5 ± 1.1 (2–3)	1.5 ± 1.1 (2–3)
*Actual anthropometrics*			
Weight, mean ± SD (IQR), kg	53.8 ± 14.8 (43–63)	50.9 ± 12.1 (43–58)	52.4 ± 13.6 (43–60)
Height, mean ± SD (IQR), cm	159.4 ± 11.6 (150–169)	157.1 ± 8.8 (152–163)	158.2 ± 10.4 (151–166)
BMI, mean ± SD (IQR), kg/m^2^	20.9 ± 4.1 (17.9–23.3)	20.4 ± 3.8 (17.8–22.5)	20.7 ± 3.9 (17.9–22.8)
BMI z-score, mean ± SD (IQR)	0.48 ± 1.04 (−0.22–1.35)	0.35 ± 0.97 (−0.28–1.02)	0.43 ± 1.02 (−0.25–1.18)
*Reported anthropometrics*			
Weight, mean ± SD (IQR), kg	51.2 ± 13.3 (41–60)	49.1 ± 11.6 (40–56)	50.1 ± 12.5 (40–58)
Height, mean ± SD (IQR), cm	159.4 ± 12.1 (150–169)	157.6 ± 9.7 (150–165)	158.5 ± 11 (150–165)
BMI, mean ± SD (IQR), kg/m^2^	19.9 ± 3.4 (17.5–21.8)	19.5 ± 3.3 (17.1–21.4)	19.7 ± 3.4 (17.1–21.4)
*Overweight/Obesity*			
Overweight, no. (%)	580 (33)	517 (29.6)	1097 (31.3)
Obese, no. (%)	230 (13.1)	194 (11.1)	424 (12.1)

^a^ ≥2000 residents. ^b^ Sum of paternal and maternal education scores: 1. for primary education (1–6 years), 2. for secondary (7–12 years), 3. for tertiary—non-university and 4. for university studies. In case of single-parent families, the education score of the parent was doubled. Score range: 2–8. BMI: body mass index, IQR: interquartile range.

## Data Availability

Not applicable.

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
