# Peer review of "Body-Size Misperception among Overweight Children and Adolescents in Greece: A Cross-Sectional Study"

_nutrients, 2023, doi:10.3390/nu15081814_

Round 1

Reviewer 1 Report

Thank you for the opportunity to review this article.

An interesting topic, with a quite well-described introduction and discussion. The chapter on statistical analyses and tables needs to be refined. English language and style need improvement (minor shortcomings are present). Greater accuracy in attention to detail is needed.

Keywords :

- Words in keywords should start with lowercase letters

Materials and Methods :

- Please reorder the subtitle to „MATERIALS AND METHODS”

- Were there any inclusion or exclusion criteria? Please specify.

- Describe the body height measurement procedure. Whether the study participant in fasting status? Were all children examined dressed the same way during the study?

- BMI z-scores and percentiles were determined according to ”international norms”, I suggest using a different term regarding international norms (the term is too general, please specify)

- Please include a description of the statistical methods used (in chapter methods and materials)

- Table 1 should be refined. I suggest changing the title to  the following: Characteristics of the study sample. It would be good to see the stats by gender. I suggest adding columns for girls and boys in addition to the "all" column. The number of overweight and obese children should also be included in the table. The characteristics of some data can be extended by the interquartile range (age, BMI z-score)

Discussion :

- Please take care of the details, for example: instead „a study by Bordeleau M. et al” should be „ a study by Bordeleau et al.”

- Please change „disagreement” to „incompatibility”, and consider replacing the following phrase with a simpler alternative: it is possible that some overweight participants provided more “socially acceptable” answers. Add an article before „Mediterranean”.

Conclusion :

- Please revise the sentence structure and grammar of this section. It could be written more succinctly

- Consider replacing the following phrase: It is important for healthcare providers to provide nutrition education and promote healthy habits

Minor comments :

- Please place bibliography references in square brackets

- Check the punctuation and articles (e.g. after „Finally” add a comma”, change the phrase „Possible limitation in our study” to A possible limitation of our study) etc.,

Author Response

We wish to thank the Reviewer for the valuable suggestions; all have been adopted, and the
Table 1 has been modified accordingly.

Reviewer 2 Report

An Interesting research to clarify the subject of body-size misperception.

A few remarks.

In Abstract, first period, I don't understand it very well. First it seems to long to me. Maybe it could be broken in order to get more accuracy.  Or (like this): "Childhood obesity can affect both physical and mental health; body-size misperception may lead to lack of motivation to make healthy changes, or to engage in unhealthy weight loss behaviors, increasing the possibility for obese children to become obese adults."

In Discussion, page 5, second paragraph, your explanation about way  overweight girls perceived more correctly their body weight while overweight boys misperceived their weight, due to social desirability, doesn't seems clear to me.

Author Response

We wish to thank the Reviewer for the valuable suggestions; all have been adopted.